# Defect-induced helicity dependent terahertz emission in Dirac semimetal PtTe$_2$ thin films

Zhongqiang Chen[1,10], Hongsong Qiu[2,10], Xinjuan Cheng[3,10], Jizhe Cui[4], Zuanming Jin[5], Da Tian[2], Xu Zhang[1], Kankan Xu[1], Ruxin Liu[1], Wei Niu[1], Liqi Zhou[6], Tianyu Qiu[7], Yequan Chen[1], Caihong Zhang[2], Xiaoxiang Xi[7], Fengqi Song[7], Rong Yu[4], Xuechao Zhai[3] ✉, Biaobing Jin[2,8] ✉, Rong Zhang[1,9] ✉ & Xuefeng Wang[1] ✉

Nonlinear transport enabled by symmetry breaking in quantum materials has aroused considerable interest in condensed matter physics and interdisciplinary electronics. However, achieving a nonlinear optical response in centrosymmetric Dirac semimetals via defect engineering has remained a challenge. Here, we observe the helicity dependent terahertz emission in Dirac semimetal PtTe$_2$ thin films via the circular photogalvanic effect under normal incidence. This is activated by a controllable out-of-plane Te-vacancy defect gradient, which we unambiguously evidence with electron ptychography. The defect gradient lowers the symmetry, which not only induces the band spin splitting but also generates the giant Berry curvature dipole responsible for the circular photogalvanic effect. We demonstrate that the THz emission can be manipulated by the Te-vacancy defect concentration. Furthermore, the temperature evolution of the THz emission features a minimum in the THz amplitude due to carrier compensation. Our work provides a universal strategy for symmetry breaking in centrosymmetric Dirac materials for efficient nonlinear transport.

The interaction of ultrafast laser with topological materials has attracted significant attention in ultrafast optoelectronics, which not only provides an efficient way to characterize the band structures and spin textures[1,2], but also serves as a control knob to dynamically induce topological phase transitions[3–5] and nonlinear optical responses[6–8]. These intriguing quantum phenomena are closely associated with the topological characteristics of linear band dispersion[6], Berry curvature[7], and their inherent symmetries[3–5]. However, centrosymmetric Dirac materials can hardly produce spontaneous photocurrents via a second-order coupling with a pulsed electrical field owing to the symmetry requirements[5,9]. Thus, it is highly desirable to develop effective approaches to engineer the material symmetry breakings, thereby generating emergent nonlinear optical phenomena.

[1]Jiangsu Provincial Key Laboratory of Advanced Photonic and Electronic Materials, State Key Laboratory of Spintronics Devices and Technologies, School of Electronic Science and Engineering, Collaborative Innovation Center of Advanced Microstructures, Nanjing University, 210093 Nanjing, China. [2]Research Institute of Superconductor Electronics, School of Electronic Science and Engineering, MOE Key Laboratory of Optoelectronic Devices and Systems with Extreme Performances, Nanjing University, 210093 Nanjing, China. [3]Department of Applied Physics, MIIT Key Laboratory of Semiconductor Microstructures and Quantum Sensing, Nanjing University of Science and Technology, 210094 Nanjing, China. [4]School of Materials Science and Engineering, Tsinghua University, 100084 Beijing, China. [5]Terahertz Technology Innovation Research Institute, Terahertz Spectrum and Imaging Technology Cooperative Innovation Center, University of Shanghai for Science and Technology, 200093 Shanghai, China. [6]College of Engineering and Applied Sciences, Nanjing University, 210093 Nanjing, China. [7]State Key Laboratory of Solid State Microstructures, School of Physics, Nanjing University, 210093 Nanjing, China. [8]Purple Mountain Laboratories, 211111 Nanjing, China. [9]Department of Physics, Xiamen University, 361005 Xiamen, China. [10]These authors contributed equally: Zhongqiang Chen, Hongsong Qiu, Xinjuan Cheng. ✉e-mail: zhaixuechao@njust.edu.cn; bbjin@nju.edu.cn; rzhang@nju.edu.cn; xfwang@nju.edu.cn

The circular photogalvanic effect (CPGE), as a second-order nonlinear effect, has been observed in a myriad of non-centrosymmetric materials/heterostructures[10–20]. The direction of the CPGE photocurrent can be switched by reversing the chirality of the light via the spin-flip transitions[10,18,19] or the nontrivial Berry curvature contribution[11,15,20]. To date, considerable efforts have been devoted to exploring multiple excitations in topological Weyl semimetals in the vicinity of Weyl nodes, which provides a static measurement of the chirality of Weyl fermions[7,21]. However, such electrical means seriously suffer from the limited time domain and spurious effects[22,23]. In this regard, polarized terahertz (THz) emission spectroscopy based on ultrafast femtosecond lasers is a powerful optical tool for generating and detecting ultrafast spin photocurrents[24,25]. Very recently, THz emission has been observed in centrosymmetric Dirac semimetal $PtSe_2$ films due to the photon drag effect[26,27] and a second nonlinearity from a structural asymmetry[28]. Remarkably, Luo et al.[5] observed the light-induced transient phase transition and hence giant photocurrent (i.e., THz emission) in bulk Dirac semimetal $ZrTe_5$. However, THz emission in Dirac semimetals with intrinsic inversion symmetry breaking has rarely been seen.

As an emerging type-II Dirac semimetal, $PtTe_2$ exhibits high conductivity, high mobility, and good air stability, providing an ideal platform to explore exotic physical properties for various applications[29–32]. Type-II Dirac semimetals feature the Lorentz variance and linear dispersion at a tilted Dirac cone, which could lead to anisotropic transport properties[29]. Strong interlayer interactions in $PtTe_2$ could also lead to the thickness-dependent semimetal-semiconductor transitions[31,32]. Recently, defect engineering has been shown to break the inversion symmetry for inducing the intriguing Rashba effect in $PtSe_2$[33,34], which offers a promising opportunity to produce nonlinear transport in centrosymmetric Dirac semimetals.

In this article, we report the defect-gradient-induced helicity-dependent THz emission in $PtTe_2$ thin films grown by a modified two-step chemical vapor deposition (CVD) method. The main Te-vacancy ($V_{Te}$) defect is created during crystal growth along the depth orientation, as evidenced by adaptive-propagator ptychography with deep-sub-angstrom resolution. The inversion-symmetry-broken $PtTe_2$ shows a remarkable CPGE based on the polarized THz emission spectroscopy. Combining first-principles calculations and defect engineering, we unambiguously demonstrate that the helicity-dependent THz emission originates from the band spin splitting and the predominantly giant Berry curvature dipole (BCD). Moreover, temperature-dependent THz emission shows the THz amplitude reaches a minimum at ~120 K due to the carrier compensation. Our findings may provide the great potential for the realization of nonlinear optical devices based on Dirac materials containing a defect gradient.

## Results

### Evidence of a defect gradient in $PtTe_2$ films revealed by electron ptychography

$PtTe_2$ is a stacked structure of quasi 2D Te-Pt-Te layers formed through the van der Waals force. It belongs to the $C_{3v}$ symmetry point group with an inversion center, which is required for stabilizing type-II Dirac nodes[35]. In this work, large-area $PtTe_2$ films with the intentionally created $V_{Te}$ defect are obtained by a controllable modified two-step CVD process (for details see "Method" section and Supplementary Note 1 and Supplementary Figs. 1 and 2)[36–38]. The tellurization of pre-deposited Pt films is considered as a diffusion process with a finite Te concentration gradient, which is controlled by the thermodynamic conditions[36]. The X-ray diffraction (XRD) patterns, Raman spectra, Raman mapping images, and high-resolution scanning transmission electron microscopy (HR-STEM) images in a high-angle annular dark-field (HAADF) mode not only show the large-area uniformity of the as-grown films, but also consistently reflect the 1$T$-phase crystal structure

of as-grown $PtTe_2$ thin films without additional impurities (see Supplementary Figs. 3-8).

With the unique deep-sub-angstrom resolution and low-electron doses, electron ptychography is a powerful tool to obtain images of single-atom defects and their depth-dependent distribution in various material systems, such as $MoS_2$, $PrScO_3$, zeolites, and $SrTiO_3$[39–42]. To image defects at the sub-nanometer precision in $PtTe_2$ films, multislice electron ptychography with adaptive-propagator (Fig. 1a) is also carried out. The original cross-sectional HR-STEM image of multilayer $PtTe_2$ films grown on sapphire ($Al_2O_3$) substrates exhibits that $d$-spacing of the periodically arranged Pt atoms is ~0.53 nm (Supplementary Fig. 8b), perfectly matching the $PtTe_2$ (001) 1$T$-phase structure. The corresponding ptychographic phase image (Supplementary Fig. 8c) is used for the examination of the vacancy defect. The enlarged phase images in four boxed, colored areas are displayed in Fig. 1b, c, respectively. The intensity contrast at different Te atomic sites clearly demonstrates the presence of $V_{Te}$, as reflected by the weaker intensity columns. We plot the intensity profile along the $c$-axis in Fig. 1d. The intensity of these Te peaks does decrease at these sites along the depth orientation. More electron ptychography observations of other slices show the consistent results (see Supplementary Figs. 9 and 10).

The inversion symmetry of the system can be broken by artificially introducing an organized defect distribution, which has been observed in van der Waals $PtSe_2$ layers[33,34]. To visualize the depth distribution of $V_{Te}$ in our $PtTe_2$ films, we perform the depth-profile mapping of a selected 29-layer $PtTe_2$ film in Te sites (Fig. 1e). The obvious difference of mapping results along the depth orientation indicates that the distribution of $V_{Te}$ is inhomogeneous. The $V_{Te}$-rich region is primarily concentrated in the midst of thin films. The relative phase intensity profile of Te/Pt in each layer shows the exact gradient-like rather than random distribution of $V_{Te}$ (Fig. 1f). Albeit the relatively large scatter, the experimental result from Layer 5 to 29 is largely consistent with density functional theoretical (DFT) calculations based on the growth dynamics (Supplementary Fig. 2d), indicating that the systematic gradient-like trend does exist in $PtTe_2$ system. The corresponding Te/Pt relative phase mapping further affirm that the $V_{Te}$ defect is mainly concentrated in the midst of films (Supplementary Fig. 11). By contrast, the electron ptychography for Te-passivated $PtTe_2$ films (i.e., post-annealing under the Te vapor to mostly eliminate $V_{Te}$) shows the uniform out-of-plane distribution of Te (Supplementary Fig. 12). Moreover, the presence of $V_{Te}$ in the as-grown $PtTe_2$ films is also revealed by the temperature-dependent Raman spectroscopy. As compared with the Te-passivated $PtTe_2$ films, the defective sample containing $V_{Te}$ has a significant blue-shift for the Raman vibrational peak of $E_g$ mode at low temperatures especially below 150 K, while the Raman peak of $A_{1g}$ mode keeps largely unshifted (Supplementary Note 2 and Supplementary Fig. 5c, d). This is due to the fact that the $E_g$ mode and $A_{1g}$ mode are associated with in-plane and out-of-plane vibrations, respectively. The existence of $V_{Te}$ has the larger impact on the phonon frequency shift of the $E_g$ mode[43]. The out-of-plane defect gradient naturally breaks the out-of-plane inversion symmetry. Meantime, the defect-gradient film can be considered as vertical collection of heterostructures composed of a myriad of $PtTe_2$ monolayers with different $V_{Te}$ defect concentrations. The discrepancy in adjacent monolayers naturally breaks the in-plane inversion symmetry (reduced from $C_{3v}$ to $C_1$). To this end, we provide the sufficient evidence of the out-of-plane defect gradient that induces in-plane symmetry breaking in $PtTe_2$ films through high spatial-resolution structural characterization.

### THz emission in symmetry-broken $PtTe_2$ films

Ultrafast-laser-induced THz emission based on second-order nonlinear effects has been recognized as a versatile probe of the symmetry breaking[44]. Figure 2a shows the schematic of the THz emission setup with a transmission configuration. The laser pulses have a duration of 100 fs at a central wavelength of 800 nm (1.55 eV) and a repetition rate

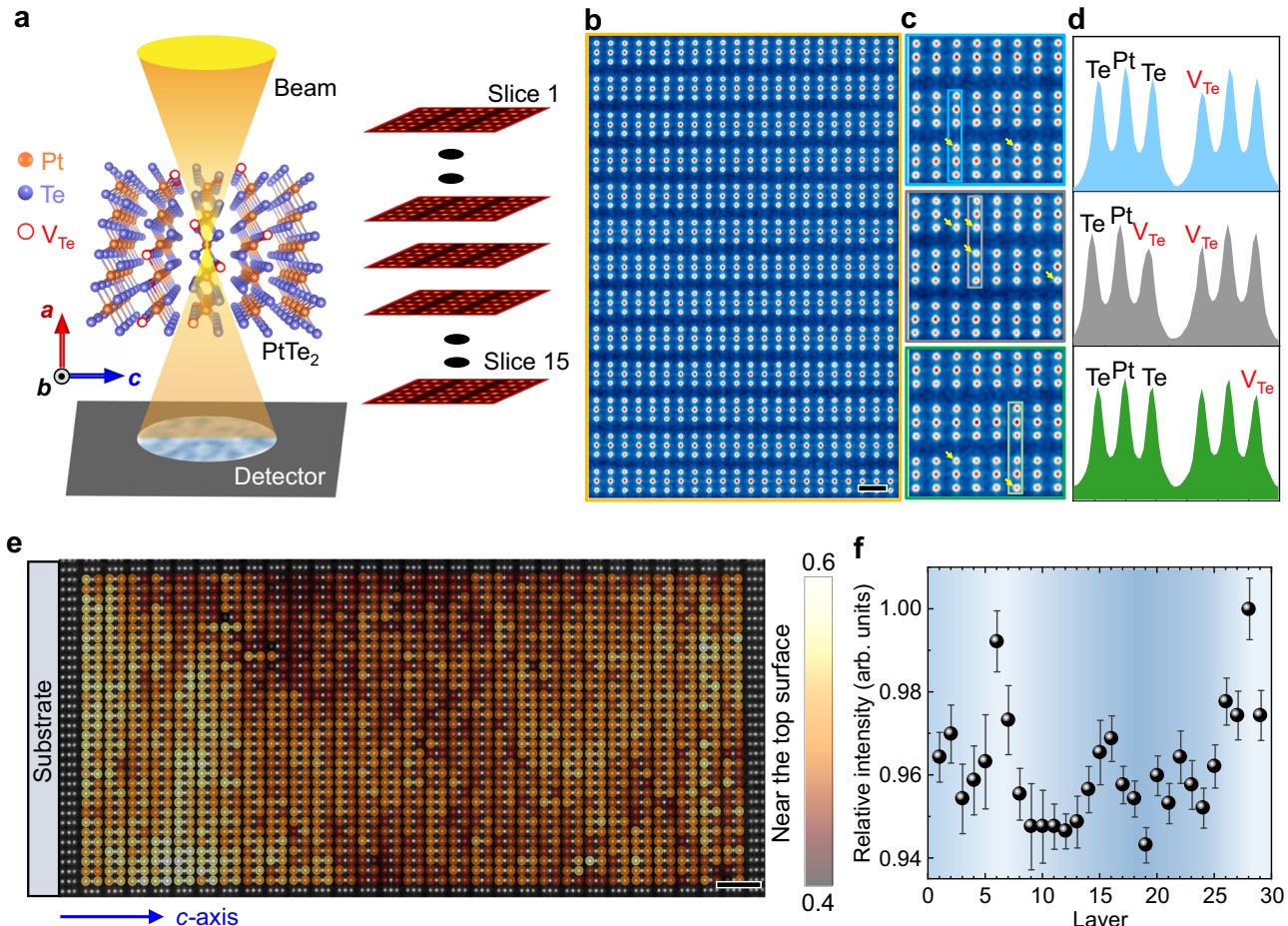

**Fig. 1 | Out-of-plane vacancy-defect gradient visualized by electron ptychography. a** Schematic illustration of the multislice electron ptychography for depth sectioning of a PtTe$_2$ sample. During ptychographic reconstruction, the sample is divided into 15 slices and each slice is ~2 nm in thickness. **b, c** The summed phase images of [001] PtTe$_2$ over 15 slices taken from orange, blue, gray, and green rectangles of Supplementary Fig. 8c, respectively. The scale bar is 5 Å. The yellow arrows marked in **c** denote the V$_{Te}$ due to a relatively weak mapping intensity. **d** The corresponding profiles of phase intensity taken from atomic columns marked with blue, gray and green rectangles in **c** respectively. **e** Te phase mapping in the

selected 29-layer PtTe$_2$ film from near the substrate to near the top surface. Light and dark colors denote the low and high V$_{Te}$ concentration, respectively. The light/dark distinction of mapping results indicates the inhomogeneous distribution of V$_{Te}$. The colorbar denotes the Te phase intensity with the unit of rad. The substrate is schematically shown. The scale bar is 1 nm. **f** The relative phase intensity variation of the Te/Pt in each PtTe$_2$ layer, which is extracted from Supplementary Fig. 11. The intensity is normalized by the maximum intensity. The error bars are generated from the standard deviations of the Te/Pt ratio.

of 1 kHz; the beam is normally incident on the PtTe$_2$ films along the z-axis. The excitation by near-infrared light enables to investigate the unique nonlinear optical properties of type-II Dirac fermions, although the observed type-II Dirac point in PtTe$_2$ resides far below the Fermi energy (~0.8 eV)[29].

To demonstrate the symmetry breaking in PtTe$_2$ films, first we investigate the THz emission under linear polarization (LP) of laser, and the various THz emission is observed under LP excitation by rotating the half-wave plate (HWP) with an angle $\phi_{\lambda/2}$ in defective PtTe$_2$ films (Fig. 2b). Note that the THz emission efficiency of our PtTe$_2$ films is three order magnitude larger than that of the standard ZnTe (Supplementary Fig. 13 and Supplementary Table 1), and it is comparable to those of topological semimetals[13,27]. The linear pump-fluence dependence of THz emission indicates that the emitted THz signals are dominated by a second-order nonlinear effect (Supplementary Fig. 14). The amplitude of THz emission exhibits a cosinoidal dependence on the $\phi_{\lambda/2}$ with a period of 180°. Such a THz radiation is derived from the linear photogalvanic effect (i.e., shift current) owing to a real-space shift of the charge centers between initial and final electron states[45]. It is noticeable that the current has a geometrical origin that can be described by the Berry phase connection[46]. In contrast, no THz signals

can be detected from the Te-passivated samples without the defect gradient (see bottom panel of Fig. 2c). Hence, the symmetry-broken PtTe$_2$ films can serve as an attractive platform to study the exotic nonlinear optical phenomena.

As a further THz emission measurement, the as-grown PtTe$_2$ films are subjected to left/right circularly polarized (LCP/RCP) excitations under normal incidence. Notably, the THz radiation and its uniformity are clearly observed in the as-grown PtTe$_2$ films (Fig. 2c and Supplementary Fig. 15). Likewise, no THz signals of helicity are detected from the Te-passivated sample (Fig. 2c). Together with the identical THz radiation intensity from the defective PtTe$_2$ films grown on other substrates of MgO and quartz (Supplementary Fig. 16), the genuine defect-gradient-induced effect is thus nailed down. The emitted THz signals are linearly polarized under the LCP or RCP excitation (Supplementary Fig. 17). The difference between the THz radiation under the LCP and RCP excitation can be attributed to a second-order nonlinear CPGE, which can also be observed in other PtTe$_2$ films with different thicknesses. The CPGE amplitude (the difference between the LCP and RCP excitations, Fig. 2d) can be modulated by the V$_{Te}$ contribution (Eq. (2)) from the thickness-dependent experiments (Supplementary Fig. 18 and Table 2). The polarity of the emitted THz signals

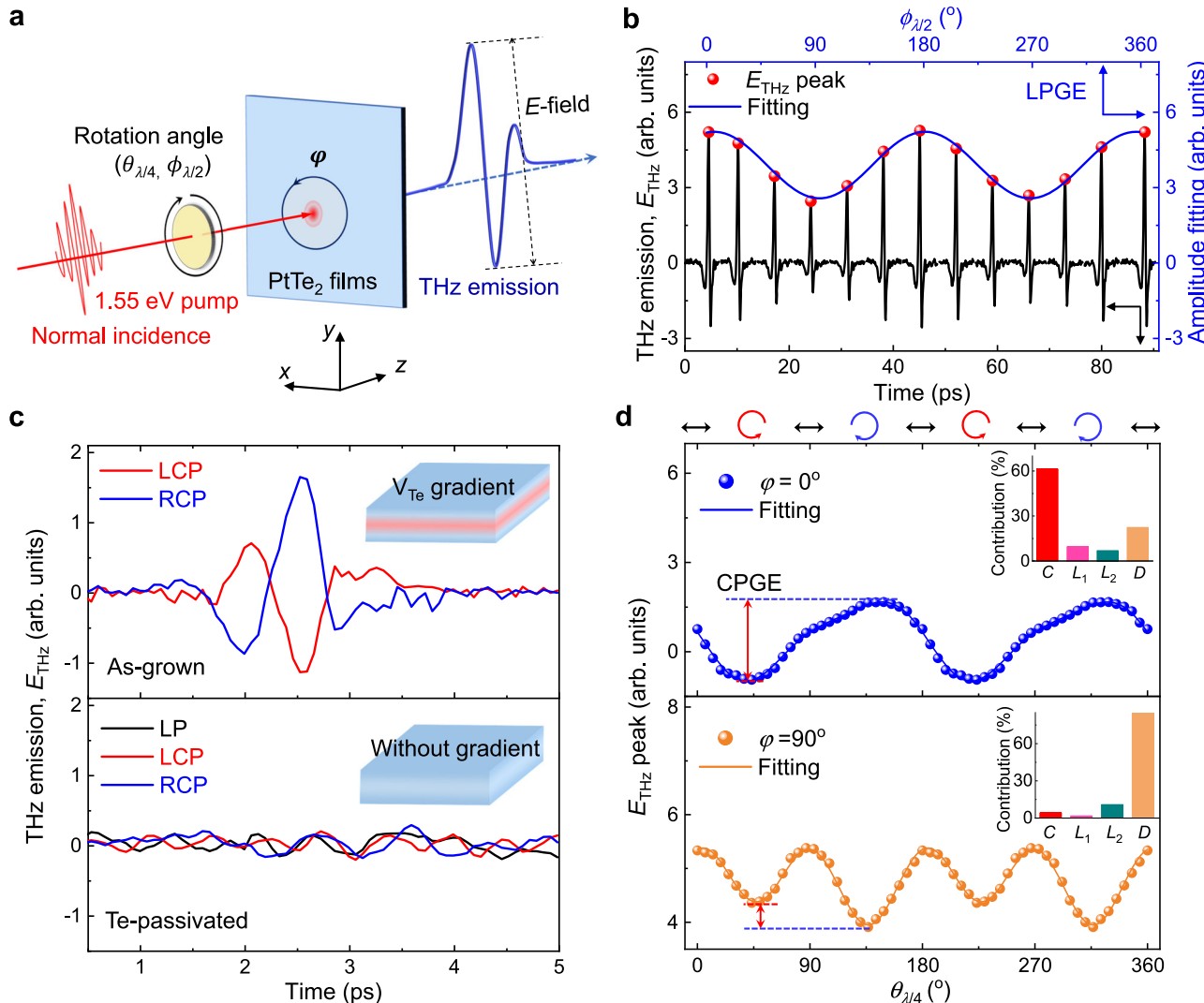

**Fig. 2 | Helicity dependent THz emission observed in symmetry-broken PtTe₂ films with the thickness of 10 nm. a** Schematic illustration of the experimental setup for THz emission measurements. The half-wave plate (HWP) and quarter-wave plate (QWP) are placed before the PtTe₂ films to change the polarization of normally incident laser pulses by varying the angles of $\phi_{\lambda/2}$ and $\theta_{\lambda/4}$, respectively. The coordinate system (*xyz*) is adapted for the laboratory frame. The sample is set in the *x-y* plane and the azimuthal angle is denoted by $\varphi$ with respect to the *y*-axis. The normal incidence is realized through the observation of the reflected light going back along its original optical path. **b** The THz amplitudes under different LP excitations. LPGE denotes linear photogalvanic effect. The black and blue arrows indicate the corresponding axis. **c** Transient THz waveforms from the as-grown PtTe₂ film (top panel) and Te-passivated PtTe₂ films (bottom panel) under the RCP

and LCP excitations at $\varphi = 0°$, respectively. The THz emission of the annealed sample under the LP excitation is also included. The inset of **c** shows the schematic diagrams of the distribution of $V_{Te}$ for the as-grown and passivated samples, respectively. **d** THz peak amplitudes at $\varphi = 0°$ (blue) and $\varphi = 90°$ (orange) as functions of $\theta_{\lambda/4}$, where the solid line represents the fitting. Arrows atop the panels indicate the polarization sequences, where the red and blue circles represent LCP and RCP, respectively, and the black double arrows represent LP. The red double arrow shows the difference between the LCP and RCP excitations, which is defined as the CPGE amplitude. The insets show the normalized fitting parameters ($C$, $L_1$, $L_2$ and $D$) extracted from fittings of the THz emission. The CPGE contributions are marked by red bars, which has the strong dependence on the $\varphi$, indicating strong in-plane anisotropy.

switches when the pump laser changes between the LCP and RCP excitation. It could be considered as an asymmetry in the transient carrier population in the momentum space, as determined by the helicity-dependent optical selection rules (Fig. 3a)[15,47]. Other mechanisms for the THz emission are excluded according to our measurement geometry (Supplementary Note 3 and Supplementary Fig. 19).

To systematically investigate the helicity-dependent THz emission, we change the helicity of the elliptically polarized light by rotating the angle of the quarter-wave plate (QWP, $\theta_{\lambda/4}$). Figure 2d shows the dependence of the THz emission amplitude on the pump laser helicity at sample azimuthal angle of $\varphi = 0°$ (top panel) and 90° (bottom panel). The laser helicity continuously changes from LCP to RCP by rotating the QWP from 0° to 360°. Under normal incidence, the polarity of the THz radiation pulse is reversed with the helicity of the

pump laser from LCP to RCP at $\varphi = 0°$, whereas the THz signals do not exhibit the polarity-reversal behavior along the other crystalline axis of $\varphi = 90°$. The THz amplitude can be connected with the photocurrent via the Maxwell equations, so we can use the photocurrent to scale the THz amplitude. The polarization dependence of the photocurrent can be fitted by a general expression[48]:

$$J = C \sin 2\theta + L_1 \sin 4\theta + L_2 \cos 4\theta + D \qquad (1)$$

where $\theta$ is the angle of the QWP, $C$ is the coefficient that determines the magnitude of the CPGE photocurrent, $L_1$ and $L_2$ represent the LP-dependent coefficients, and $D$ is the polarization-independent background that arises from the thermal effect owing to the light absorption. Parameters $C$, $L_1$, $L_2$, and $D$ all have finite contributions,

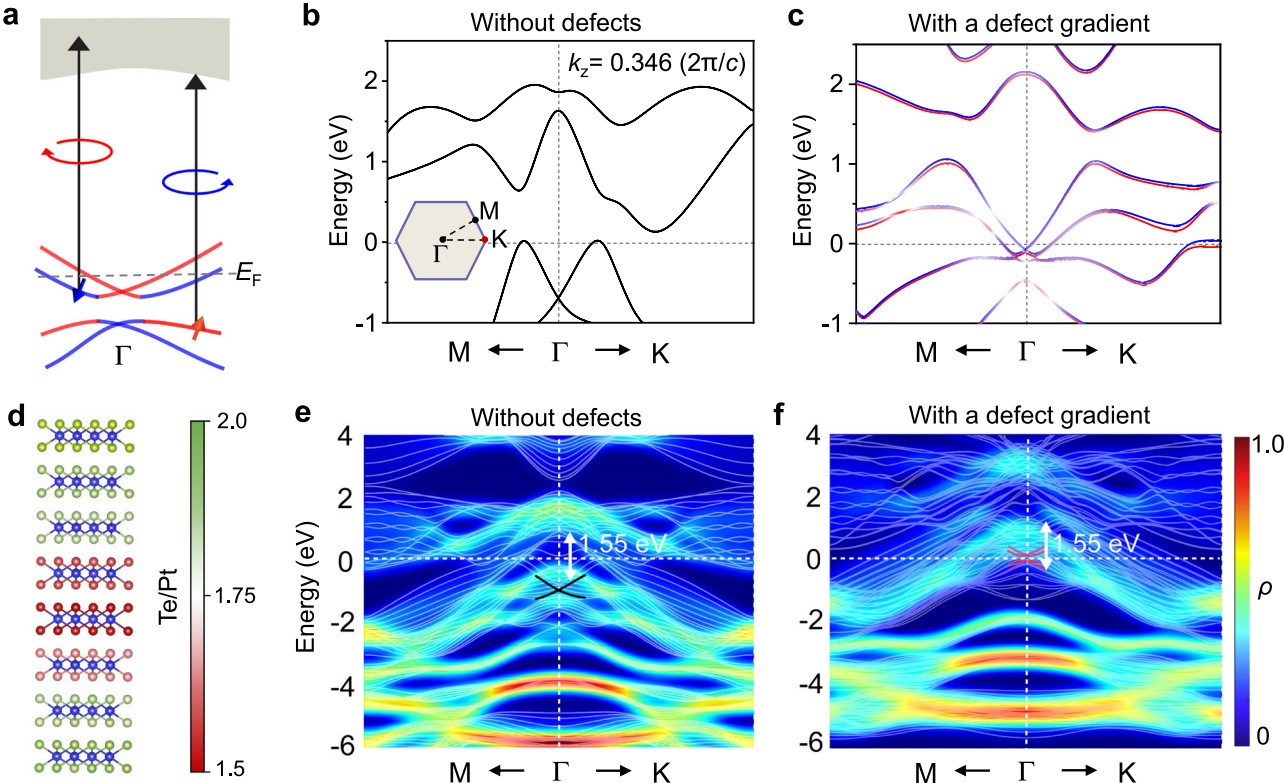

**Fig. 3 | Band spin splitting enabled by a defect gradient. a** The band diagram of the optical selection rules for the symmetry-broken PtTe$_2$. The dashed line denotes the Fermi level ($E_F$). The blue and red arrows denote the electronic states with the opposite spins. **b, c** Band structures for the bulk samples without defects and with a defect gradient, respectively. The high-symmetry points Γ, M, and K in the ($k_x$, $k_y$) plane of the reciprocal space are shown in the inset of **b. d** Schematic diagram of an 8-layer-thick PtTe$_2$ film with a defect gradient. The V$_{Te}$ gradient is indicated by the color scheme on the right. **e, f** Band structures for a concrete 8-layer-thick PtTe$_2$ sample without defects and with a defect gradient, respectively. Notably, spin degeneracy is preserved in both defect-free bulk in **b** and defect-free 8-layer-thick PtTe$_2$ sample in **e**. However, a band spin splitting occurs in **c, f** where the number of subbands increases. The colors denote the proportion $\rho$ contributed by the upper and lower surface layers. The white double arrows schematically show the electron transition range under 800-nm excitation (-1.55 eV) near the Dirac point. The black and red crossing curves in **e, f** denote the Dirac point and spin-splitting bands, respectively.

while $C$ is dominant along the rotational asymmetric direction of $\varphi = 0°$. On the contrary, at $\varphi = 90°$, polarization-independent $D$ dominates, while the others are negligible. The fitting details are presented in the insets of Fig. 2d, indicating that the observed CPGE photocurrents have a strong in-plane anisotropy. It is noticeable that $C$ shows a significant dependence on the sample azimuthal angle $\varphi$ due to the lowered symmetry (Supplementary Fig. 20), which is distinct from $\varphi$-independent CPGE observed in materials with the $C_3$ rotational symmetry[49]. The detailed symmetry analysis further proves that symmetry point group of defect-gradient PtTe$_2$ system is lowered from $C_{3v}$ to $C_1$ (Supplementary Note 4 and Supplementary Fig. 21). The modulation of the CPGE with the helicity of the laser pump shows that the ultrafast spin photocurrent is more pronounced along the inversion-symmetry axis (i.e., $\varphi = 0°$) for opto-spintronic applications. As a complementary tool to the THz emission, the optical second harmonic generation is further measured to confirm the inversion symmetry breaking in defect-gradient PtTe$_2$ films (Supplementary Fig. 22).

### First-principles calculations for band structures of PtTe$_2$
As mentioned above, the observed helicity-dependent THz radiation is attributed to the optical selection rule. A spin splitting arising from the intrinsically broken inversion symmetry and the strong spin-orbit coupling should occur in the defect-gradient PtTe$_2$ (schematically shown in Fig. 3a). To elucidate the influence of the V$_{Te}$ gradient on the PtTe$_2$ band structure for the helicity-dependent THz emission, DFT-based first-principles calculations are performed. The band

structure of the bulk defect-free PtTe$_2$ sample near the Dirac point shows no spin splitting at all because the system has the spatial inversion symmetry (Fig. 3b). To understand the CPGE induced by the V$_{Te}$ defect gradient in the experiment, we use the average defect gradient between two Te sub-lattices within a minimum approximation model of the bulk PtTe$_2$ sample, which roughly capture the key band variation (Fig. 3c). Compared to the band structure of PtTe$_2$ without defects (Fig. 3b), a considerable gap does open near the Dirac point and the momentum-dependent spin splitting happens. It can be further verified from the different directions (see Supplementary Fig. 23 and Fig. 4e, f). The band spin splitting energy ($\Delta = 15 \pm 5$ meV at Γ point and $\Delta = 70 \pm 5$ meV at K point) is comparable to the other typical materials (note that $\Delta$ at K point is larger than those of Weyl semimetals[16,50], Supplementary Table 3). The upshift of the Dirac point, closer to the Fermi level, is more favorable for the photoexcitation. Furthermore, we also compare the band structure in samples with the uniform defects and a defect gradient along the depth (Supplementary Fig. 24), and find that the presence of the defect gradient is critical to achieve the selective excitation with respect to the uniform defects.

A previous experiment demonstrated that PtTe$_2$ films with more than three layers exhibited the bulk band features[32]. To precisely explore the V$_{Te}$ gradient in the as-grown films, we further choose an 8-layer-thick PtTe$_2$ sample (corresponds to the thickness of 4 nm with the THz emission shown in Supplementary Fig. 18a) and provide the DFT-calculated band structures with and without a defect gradient. A slab geometry composed of 8 layers of PtTe$_2$ and 20 Å vacuum is adopted to describe the genuine experimental situation (Fig. 3d).

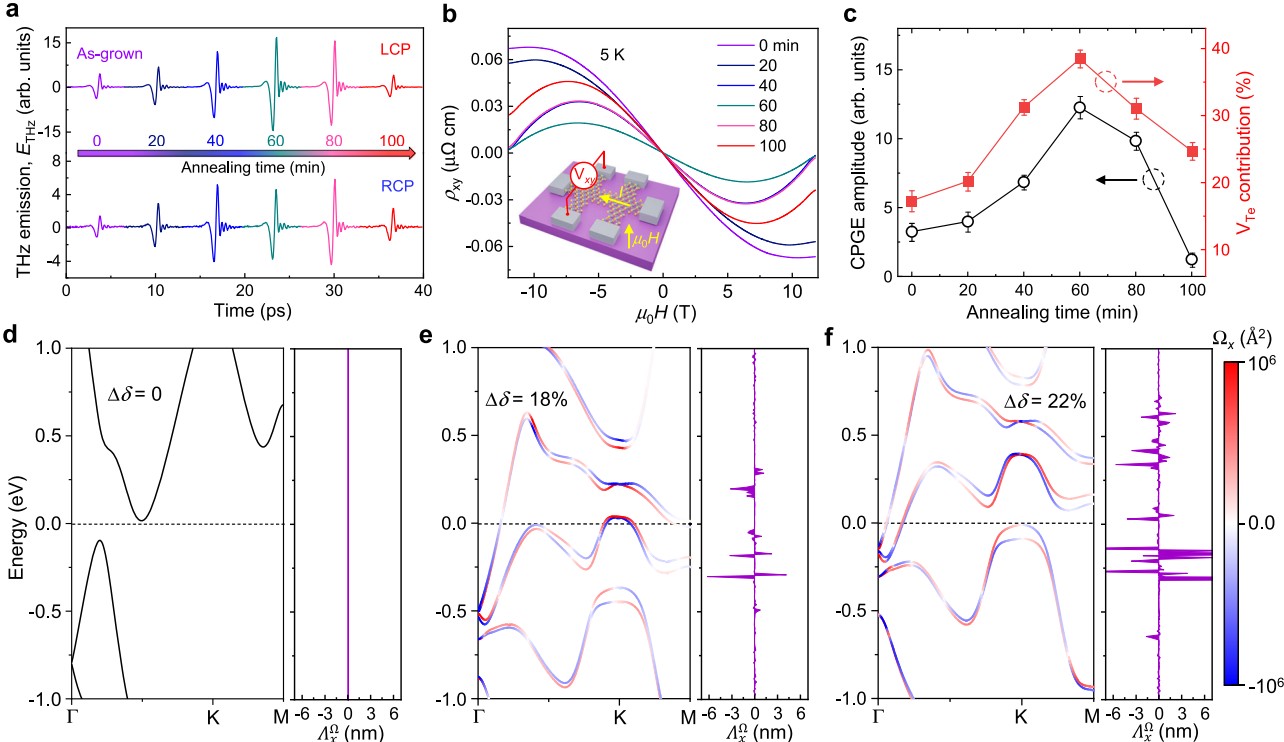

**Fig. 4 | Microscopic origin of THz emission through defect engineering. a** The transient THz waveforms of PtTe$_2$ films with various *in-vacuo* annealing time under LCP and RCP at $\varphi = 0°$, respectively. **b** Hall curves of PtTe$_2$ films with various *in-vacuo* annealing time under the perpendicular field at 5 K. The inset shows a schematic of Hall bar device structure. **c** The extracted CPGE amplitude from **a** and extracted V$_{Te}$ contribution from **b** as a function of the annealing time. The lines are drawn as guides for the eyes. The error bars of the CPGE amplitude and the V$_{Te}$ contribution indicate the uncertainties from the calculations of the difference between LCP and RCP and the fitting of the experimental results, respectively. The thickness of PtTe$_2$ films is 18 nm. **d–f** Band structures and Berry curvatures of PtTe$_2$ obtained within the home-made codes. DFT bands are fitted for the defect-free sample ($\Delta\delta = 0$) in **d** and the defective samples with V$_{Te}$ concentration difference $\Delta\delta = 18\%$ in **e** and $\Delta\delta = 22\%$ in **f** between two Te atoms in the minimum model, respectively. The right panels in **d–f** show the $x$ component of BCD ($\Lambda_x^\Omega$) for the defect-free and defective samples, respectively. No BCD is found in the defect-free sample.

Notably, spin degeneracy is preserved in the stoichiometric sample because the inversion symmetry is protected even by decreasing the thickness of the film from bulk to 8 layers (or any other number of layers) (Fig. 3e). Very remarkably, spin splitting does occur in the V$_{Te}$-gradient sample containing a larger number of subbands owing to the inversion symmetry breaking (Fig. 3f). It is seen that band structures are quite different between defect-free and defective PtTe$_2$, leading to the dramatic changes of the band structure of the defect-free material. These theoretical results further consolidate that the spin splitting induced by the symmetry breaking is a prerequisite for the experimentally observed emergent THz emission.

**Defect engineering of the helicity-dependent THz emission**

As shown above, the V$_{Te}$ defect gradient breaks both the $C_3$ rotational symmetry and the inversion symmetry, which may lead to the asymmetric distribution of Berry curvature ($\mathbf{\Omega(k)}$) and non-vanishing BCD ($\mathbf{\Lambda^\Omega}$), namely the dipole moment of Berry curvature over the occupied states[51]. Berry curvature is an important geometrical property of Bloch bands, which plays a dominant role in the nonlinear transport in time-reversal-invariant materials[52–54]. It has been theoretically shown that the CPGE is directly proportional to the BCD[47,55–57].

We investigate the dependence of the CPGE amplitude extracted from THz emission on the tunable V$_{Te}$-defect concentration in 18-nm-thick PtTe$_2$ films under in-vacuo annealing. The V$_{Te}$ formation is thermodynamically favorable since it has a lower formation energy under a Te-poor annealing environment. As seen from the annealing-time-dependent THz emission (Fig. 4a), we find that the THz emission intensity first increases until it reaches a maximum ~1 h of annealing, and then decreases with time evolution. The micro-Raman spectra taken from all annealed films ensure the invariable 1T phase structure during the *in-vacuo* annealing (Supplementary Fig. 25). In view of the band structure tunable by the defect concentration[58], we accordingly carry out the Hall measurements at 5 K. The Hall signals of all annealed films show a negative slope and a nonlinear characteristic (Fig. 4b), in sharp contrast to the typically linear Hall behavior of Te-passivated PtTe$_2$ films at 5 K (Supplementary Fig. 18d). We simultaneously fit the Hall and longitudinal magnetoconductance using a two-band model to extract the carrier concentration and mobilities (Supplementary Note 5 and Supplementary Table 4). Since the carrier compensation occurs below 120 K (as discussed below) we fix the equal carrier density during fitting (Supplementary Table 4). Because the incorporated V$_{Te}$ in PtTe$_2$ arouses the nonlinear characteristic of Hall curves, which contributes to the hole carrier conduction, we use the V$_{Te}$ contribution expression as the quantification of the effective vacancy defect.

$$V_{Te}\text{contribution} = \frac{n_h\mu_h}{n_e\mu_e + n_h\mu_h} \times 100\% \qquad (2)$$

Very strikingly, the extracted CPGE amplitude and V$_{Te}$ contribution show the very similar time evolution (Fig. 4c). The PtTe$_2$ films annealed for one hour exhibits the largest CPGE amplitude with the maximum V$_{Te}$ contribution.

Next, we turn to unveil the microscopic mechanism of the helicity-dependent THz emission (namely CPGE) using the BCD theory. The

band structures of PtTe$_2$ with/without the V$_{Te}$ defect gradient in other directions of Brillouin zone (BZ) further confirm the apparent band spin splitting in defective samples (Fig. 4d–f). Moreover, the Berry curvatures in the subbands increase with increasing defect gradient. It is noted that the CPGE is strongly dependent on the V$_{Te}$ contribution, thus the Berry curvatures serve as the crucial evidence of the physical picture behind the CPGE (for details see Supplementary Note 6). In brief, the BCD can be calculated by[51,52,59]:

$$\Lambda_\alpha^\Omega(\epsilon_{\mathbf{k}}) = \sum_n \int_{BZ} \frac{d\mathbf{k}}{(2\pi)^2} \Omega_z^n \frac{\partial \epsilon_{\mathbf{k}}^n}{\hbar \partial k_\alpha} \frac{\partial f(\epsilon_{\mathbf{k}}^n)}{\partial \epsilon_{\mathbf{k}}^n} \tag{3}$$

where $\alpha$ denotes the spatial index $(x, y)$, $\epsilon_{\mathbf{k}}^n$ is the $n$th-band energy, $f(\epsilon_{\mathbf{k}}^n)$ represents the Fermi-Dirac function, and a sum over all the subbands crossing a fixed energy is included in the calculations. As $\Lambda_\alpha^\Omega$ is obtained, the CPGE photocurrent under normal incidence can be determined by ref. 11:

$$\mathbf{J}^{CPGE} = \frac{e^3\tau}{\pi\hbar^2} \text{Im}[\mathbf{E}(-\omega) \times \hat{c}(\Lambda^\Omega \cdot \mathbf{E}(\omega))] \tag{4}$$

where $\tau$ is the carrier relaxation time and $\mathbf{E}(\omega) = \frac{E_0}{\sqrt{2}}(e^{i\omega t}, e^{i(\omega t \pm \frac{\pi}{2})}, 0)$ describes the electric field of the normal incident RCP and LCP light. As long as the BCD is nonzero (i.e., $\Lambda_\alpha^\Omega \neq 0$), the CPGE photocurrent can be observed under normal incidence in the symmetry-broken PtTe$_2$. In right panels of Fig. 4d–f, it is seen that the BCD ($\Lambda_\alpha^\Omega$) becomes more dominant when the V$_{Te}$ gradient is larger. The larger V$_{Te}$ gradient, the more V$_{Te}$ contribution. Hence, the BCD's trend is entirely consistent with our experimental observation (Fig. 4c). Notably, the BCD value is giant, which can reach the order of 10 nm with a maximum value up to ~64 nm (Supplementary Fig. 26a), typically larger than that (-8 Å) calculated in monolayer WTe$_2$[11]. Importantly, the carrier relaxation time ($\tau$) will become short once the V$_{Te}$ concentration is excessively high upon the elongation of the annealing time. Consequently, the CPGE amplitude drops after 1-h annealing and reaches a minimum value according to Eq. (4). We thus obtain the quantitative relationship between the CPGE and the BCD through the defect engineering.

### Temperature-dependent THz emission in defect-gradient PtTe$_2$ films

It should be noted that the contribution of defects becomes more pronounced as the temperature decreases, as manifested by the temperature-dependent Raman spectra (Supplementary Fig. 5). Therefore, it is highly desirable to reveal the influence of defects on THz radiation at low temperatures. The 35-nm-thick PtTe$_2$ films were cryogenically cooled during measurements. Intriguingly, with decreasing temperature, both THz amplitudes under LCP and RCP in principle decrease monotonously and then increase (Fig. 5b), featuring a minimum near 120 K (light-purple-shaded area). Given that the carrier compensation effect destroys the spin photocurrent generation[50], we then measure the temperature-dependent Hall behaviors of the PtTe$_2$ films (Fig. 5c). Above 120 K, the Hall coefficient is negative, suggesting the electron-type carriers dominate the transport behavior. As the temperature decreases, the Hall coefficient remains negative at low fields, but the slope decreases at the higher fields, indicating the presence of multiband effects in PtTe$_2$. The temperature dependence of the carrier density and mobility is plotted in Fig. 5d. The extracted carrier density shows an electron-rich case at high temperatures and a nearly compensated carrier concentration below 120 K, which exactly coincides with the minimum of the THz amplitudes (Fig. 5b). As temperature reaches 120 K, the excited electrons and holes compensate each other, resulting in a diminished photocurrent. When temperature is lower than 120 K, the increasing mobility of electrons leads to an increase in

photocurrent, although carrier compensation is still maintained. When temperature is above 120 K, the carrier compensation gradually decreases, and electron carriers increase dominantly (the inset of Fig. 5d), thus enhancing the photocurrent. Consequently, the minimum of the THz amplitude observed near 120 K can be attributed to the electron-hole compensation. This provides a profound insight into the temperature-dependent THz emission via V$_{Te}$-induced Hall nonlinear characteristic. It should be noted that albeit the strong THz emission amplitude, the CPGE amplitude is relatively weak, i.e., the helicity-dependent THz emission is not remarkable from 50 to 300 K (Fig. 5b), which is attributed to the fact that when the film is thick it is difficult to control V$_{Te}$ gradient-like distribution throughout the depth orientation. The relatively prominent CPGE amplitude at 50 K is attributed to the more V$_{Te}$ contribution at low temperatures (Supplementary Fig. 5).

## Discussion

We have demonstrated the generation of vacancy-defect-gradient-induced helicity-dependent THz emission via CPGE in PtTe$_2$ thin films. The high spatial resolution of adaptive-propagator ptychography enables us not only to directly observe the sub-angstrom V$_{Te}$, but also to provide its depth distribution. These results affirm our claim that the out-of-plane vacancy gradient breaks the inversion symmetry, inducing the considerable band spin splitting and the anisotropic CPGE photocurrent. The CPGE is activated by the giant BCD due to the vacancy-defect gradient along the depth orientation, as corroborated by the theoretical calculations. The CPGE amplitude can be largely tunable by the defect concentration using the in-vacuo annealing. The temperature evolution of the helicity-dependent THz emission reveals that the minimum of the THz emission amplitude is attributed to the compensated electron and hole carriers. Furthermore, the helicity-dependent THz emission is also observed in the Se-vacancy-gradient PtSe$_2$ films under normal incidence, indicating the universality of our unique strategy of the symmetry engineering in Dirac semimetals (Supplementary Fig. 27). Our work provides an alternative pathway to manipulate spin photocurrents that are otherwise unavailable in centrosymmetric Dirac semimetals, which facilitates the development of nonlinear optical devices and integrated THz spintronic devices based on quantum materials.

## Methods

### Growth of PtTe$_2$ films

Large-area PtTe$_2$ films with the various thickness were grown by a modified two-step CVD process on the (0001)-oriented Al$_2$O$_3$ substrate (Shanghai Institute of Optics and Fine Mechanics). First, Pt films with controlled thickness of 1–10 nm were deposited on clean Al$_2$O$_3$ substrates by magnetron sputtering at a fixed rate of 1.25 Å s$^{-1}$. Second, the pre-deposited Pt films were tellurized to form large-area and high-quality PtTe$_2$ films (for details see Supplementary Note 1). Te powder, as a reaction source, was placed in a quartz tube with an inner diameter of 10 mm and two open ends. In addition, the furnace was heated to a growth temperature of 400 °C at a rate of 13.3 °C min$^{-1}$ and maintained for 30–120 min, followed by a forming gas (95% Ar and 5% H$_2$) delivered at a rate of 100 standard cubic centimeters per minute. Subsequently, it was cooled down to room temperature naturally after the growth. During the growth process, the carrier gas was maintained to prevent the samples from oxidation. Control samples were prepared by post annealing under Te vapor to mostly eliminate the V$_{Te}$ defect gradient, which is regarded as Te-passivated. The PtTe$_2$ films grown on other substrates of MgO (111) and quartz glass were also prepared by the same procedure. The in-vacuo annealed samples with thickness of 18 nm were heated to 300 °C for duration from 20 to 100 mins, with the flowing forming gas applied to accelerate the V$_{Te}$ formation.

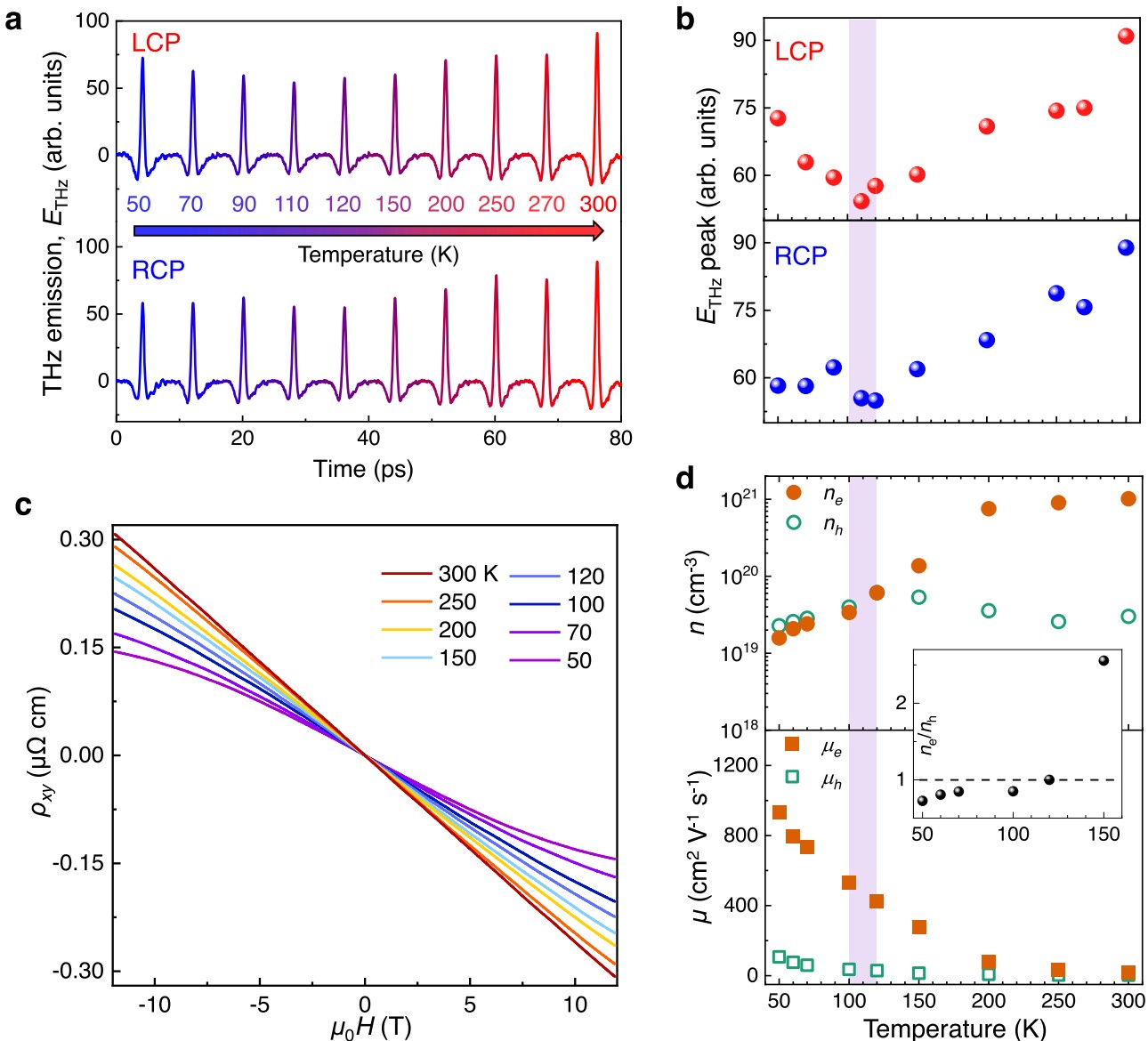

**Fig. 5 | Temperature-dependent THz emission and its correlation with carrier compensation. a** Temperature-dependent transient THz waveforms at $\varphi = 0°$ under LCP and RCP, as shown in the top and bottom panel, respectively. **b** The THz amplitudes as a function of temperature, which is extracted from **a**. **c** The temperature-dependent Hall curves of $PtTe_2$ films under the perpendicular field.

**d** The extracted carrier concentration and mobility at various temperatures. Note that the $PtTe_2$ film shows the Hall transition from nonlinearity to linearity within the temperature of the light purple region (~120 K), coinciding with the minimum of THz amplitude shown in **b**. The inset of **d** shows the temperature dependence of $n_e/n_h$ ratio. The thickness of $PtTe_2$ films is 35 nm.

## Structural characterization

The thickness of films was determined by atomic force microscopy (NT-MDT). The crystalline structure was characterized by $\theta-2\theta$ XRD using a Cu $K_\alpha$ line (Bruker D8 Discover). Micro-Raman spectroscopy was carried out by a home-made confocal microscopy in the back-scattering geometry under a 532 nm laser excitation. The low-temperature Raman spectra were collected by grating spectrograph and a liquid-nitrogen-cooled charge-coupled device. Temperature control was achieved by using a Montana Instrument Cryostation. The STEM-HAADF images and the 4D datasets used for adaptive-propagator ptychography reconstruction were acquired using the Titan Themis G2, which was equipped with a probe corrector and an image corrector. The 4D datasets were obtained using a pixel array detector known as EMPAD. The microscope operates at a high tension of 300 kV and a convergence semi-angle of 25 mrad. The defocus value, which was positive for underfocus, was 20 nm, and the scan step

size was 0.37 Å. The diffraction patterns had a pixel size of 0.033 Å$^{-1}$ and dimensions of 128 × 128. During the reconstruction process, the diffraction patterns were padded to 256 × 256 to achieve a real-space pixel size of 0.11 Å. The detailed reconstruction process can be found elsewhere[41].

## THz emission measurements

Femtosecond laser pulses were generated by using a Ti:sapphire regenerative amplifier laser (duration of 100 fs, repetition rate of 1 kHz, and central wavelength of 800 nm). The $PtTe_2$ films were excited by the pump beam with the fluence of 0.16 mJ cm$^{-2}$ under normal incidence (Fig. 2a). The laser-induced THz signal was detected via electro-optical sampling. The pump laser pulses were loosely focused on the samples with a beam diameter of approximately 2 mm. The HWP or QWP was placed before the sample to vary the polarization state of the pump laser. We used two parabolic mirrors with a reflected focal

length of 5.08 cm to collect and refocus the emitted THz wave. We temporally probed the THz electric field by measuring the ellipticity modulation of the probe beam in a (110)-oriented ZnTe crystal with the thickness of 1 mm. All measurements were carried out in dry air (humidity less than 3%) at room temperature. Additionally, the low-temperature THz emission measurements were conducted only on the 35-nm-thick $PtTe_2$ films.

## Magnetotransport measurements

Prior to transport measurements, the Hall-bar contacts with eight-probe configuration were fabricated by silver paste on films, as schematically shown in the inset of Fig. 4b. Magnetotransport measurements were measured using a Cryogenic cryogen-free measurement system (CFMS-12). The perpendicular magnetic field up to 12 T was applied to the film surface.

## Second harmonic generation measurements

The second harmonic generation measurements were performed using a Ti:sapphire oscillator (duration: 70 fs, repetition rate: 80 MHz, central wavelength: 810 nm). The laser pulse was focused to a spot size of ~1 μm on the films by a 40× objective lens. The generated second harmonic light was detected by a photomultiplier tube, and the harmonic generation signal was collected in the reflection mode.

## First-principles DFT calculations

All the band structures were calculated using the Vienna Ab-Initio Simulation Package (VASP)[60,61] within DFT in the regime of local density approximation (LDA) implemented by a projector-augmented wave. Spin-orbit coupling was included in all calculations. We used lattice constants $a = b = 4.03$ Å and $c = 5.22$ Å for the $PtTe_2$ crystal[31], and test $k$-point meshes and energy cutoff for the convergence to stabilize the structure. The plane wave basis energy cutoff and the force convergence standard were set to 500 eV and 0.01 eV Å$^{-1}$, respectively. The self-consistent convergence criterion was set to $10^{-6}$ eV. Here, we mainly showed the DFT band structures of the bulk $PtTe_2$ sample and 8-layer thick sample, where the $8 \times 8 \times 6$ grid of $k$-points was used for the former and the $13 \times 13 \times 1$ grid point sampling with Γ-centered Monkhorst-Pack adopted for the latter. To eliminate the interaction of adjacent slab calculations, vacuum spacing (16 Å) was added above the topmost layer of the 8-layer thick sample. To simulate the $V_{Te}$ of the experimental samples, we employed the virtual crystal approximation (VCA) method[62], which was proven to be valid in studying the topological phase transition[63] and the Rashba physics[34]. Average defects within a minimum approximation model were used to roughly capture the key band variation, where the vacancy defect gradient was set between two Te sub-lattices. In addition, we further used the widely-used open source Wannier90 codes to fit the DFT band structures. Meanwhile, we developed the home-made codes to obtain the corresponding effective Hamiltonian and then calculated the subband-dependent Berry curvature and the energy-dependent BCD (see Supplementary Note 6 for details).

## Data availability

Relevant data generated in this study are provided in the article and Supplementary Information file. All raw data that support the plots within this paper and other findings of this study are available from the corresponding authors upon request. Source data are provided with this paper.

## Code availability

The code that supports the theoretical plots within this paper is available from the corresponding authors upon request.

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

## Acknowledgements

This work was supported by the National Key Research and Development Program of China (grant no. 2022YFA1402404 to X.W.; grant no. 2023YFF0719200 to Z.J.), the National Natural Science Foundation of China (grant nos. T2394473, T2394470, 62274085, 11874203, and 61822403 to X.W.; grant nos. 62374088 and 12074193 to X.-C.Z.; grant nos. 92161201 and 12025404 to F.S.; grant no. 62322115 to Z.J.; grant no. 62027807 to B.J.), and the Fundamental Research Funds for the Central Universities (grant no. 021014380080 to X.W.; grant no. 021014380176 to H.Q.). The authors acknowledge Hongming Weng, Jingbo Qi, Dong Sun, Ya Bai, and Guozhong Xing for valuable discussions.

## Author contributions

X.W. conceived the study and proposed the strategy. X.W. and R.Z. supervised the project. Z.C., X.Z. and K.X. developed the CVD method, grew the samples, and performed XRD measurements. H.Q., Z.J. and D.T. carried out the THz emission measurements. X.C. and X.-C.Z. conducted theoretical calculations. Z.C., X.-C.Z. and X.W. did symmetry analysis. J.C., L.Z. and R.Y. carried out the microscopic characterization. Z.C., X.Z., R.L. and Y.C. fabricated the devices and performed transport measurements. Z.C., T.Q. and X.X. performed Raman and second harmonic generation measurements. W.N., C.Z., F.S., B.J. and R.Z. contributed to the data analysis and discussion. X.W. and Z.C. wrote the manuscript with input from all the authors. All authors reviewed and commented on the manuscript.

## Competing interests

The authors declare no competing interests.
