## [Peer Review File · Nature Communications]

Defect-induced helicity dependent terahertz emission in Dirac semimetal PtTe₂ thin filmsReviewers' Comments:

Reviewer #1:

Remarks to the Author:

I looked at the referee reports, as well as the authors' reply to the referees' comments. Most comments were sufficiently rebutted. I just have three comments below.

(1) The authors claimed that their Fig. R1(a) and R1(b) are similar. They are not: (a) R1a has a very strong peak at layer 6, while R1b has peak at layer 0, (b) R1a has a smaller peak at layer 16, while R1b has no peak there. These peaks are significant because they exceed the noise level given by the error bars.

I know that sample growth is not easy at all, and so data may not agree with theory. However, the two peaks in R1a are *qualitatively*, not just quantitatively, different from R1b.

If the authors can show similar experimental figures like R1a, but for the 8-layer and 12-layer PtTe₂, then compare with the DFT figures in Fig. S2b and Fig. S2c, and show that experimental data is similar with theory (DFT), then it will be much more convincing.

(2) In the *old* Fig. R6, the $\alpha = 0$ deg (normal incidence) THz amplitude is smaller than that of $\alpha = \pm 15$ deg. However, in the new Fig. R4, the $\alpha = 0$ deg THz amplitude is now similar or larger than that of ± 15 deg. Can the authors explain this?

(3) Related to point (2) above. The authors are correct to note that when the incident angle is reversed, the emitted THz does not change polarity, implying that it *cannot* be PDE. However, if the THz emission mechanism is not PDE, then one should expect that THz emission at normal incidence should be larger than at oblique incidence, which is *not* what the data shows. The authors did not address this point in their reply.

Reviewer #2:

Remarks to the Author:

Authors replied to my comments well, and I found the manuscript is now revised satisfactorily. I therefore recommend the publication of this work in Nature Communications.

RESPONSE TO REVIEWERS' COMMENTS (NCOMMS-24-07569-T)

Reviewer #1 (Remarks to the Author):

I looked at the referee reports, as well as the authors' reply to the referees' comments. Most comments were sufficiently rebutted. I just have three comments below.

Reply: Thank you very much for your valuable inputs on our work. We have carefully clarified your remaining three thoughtful comments below.

Comment (1) *The authors claimed that their Fig. R1(a) and R1(b) are similar. They are not: (a) R1a has a very strong peak at layer 6, while R1b has peak at layer 0, (b) R1a has a smaller peak at layer 16, while R1b has no peak there. These peaks are significant because they exceed the noise level given by the error bars. I know that sample growth is not easy at all, and so data may not agree with theory. However, the two peaks in R1a are *qualitatively*, not just quantitatively, different from R1b. If the authors can show similar experimental figures like R1a, but for the 8-layer and 12-layer PtTe₂, then compare with the DFT figures in Fig. S2b and Fig. S2c, and show that experimental data is similar with theory (DFT), then it will be much more convincing.*

Reply: Thank you for this insightful comment. For the slight discrepancy between the DFT theory and the experimental data, the main reason lies at: The DFT calculations have shown that the formation energy of Te vacancy in the middle region is larger than that in the surface of the PtTe₂ films, which results in a Te-vacancy gradient in the PtTe₂ films from the surface to the middle region. This trend is *qualitatively* coincident from both the theory and experiment. However, the presence of the substrate, the reaction temperature, time, and the carrier gas flow rate in the real situation should all play a critical role in the actual defect distribution in as-grown PtTe₂ films, which makes the layer-dependent Te-vacancy statistical distribution be slightly deviated from the theoretical calculation. This important information had been provided in **Supplementary Note 1** as well as in our previous rebuttal letter (**Response 1**). Albeit the relatively large scatter in experiment, the result from Layer 5 to 29 is still *largely* consistent with the theory, indicating that the systematic gradient-like trend does exist in PtTe₂ system. We are grateful for your recognition that “*I know that sample growth is not easy at all, and so data may not agree with theory.*”. Please note that we had toned down the description from “*similar*” to “*largely consistent*”.

Moreover, as seen from a myriad of research works involving 4D STEM (e.g., *Nature* **559**, 343 (2018); *Science* **380**, 633 (2023); *Science* **383**, 212 (2024)), it is conventional to perform the representative microscopic characterization on only one sample. Thus, it is unnecessary to carry out the 4D-STEM for 8-layer and 12-layer PtTe₂. Furthermore, there would be more errors in the statistical values because the fewer-layer samples should have the principal impact from the substrate. Hence, the 4D-STEM involving our state-of-the-art electron ptychography technique (developed from our coauthor Prof. Rong Yu's previous works, such as *Sci. Adv.* **8**, eabn2275 (2022); *Sci.*

Adv. **9**, eadf1151 (2023), *Nat. Commun.* **14**, 162 (2023); *Nat. Nanotechnol.* doi: 10.1038/s41565-023-01595-w) is performed on the representative 29-layer PtTe₂ film, which convincingly evidences the Te-vacancy defect gradient.

Comment (2) *In the *old* Fig. R6, the $\alpha = 0$ deg (normal incidence) THz amplitude is smaller than that of $\alpha = \pm 15$ deg. However, in the new Fig. R4, the $\alpha = 0$ deg THz amplitude is now similar or larger than that of ± 15 deg. Can the authors explain this?*

Reply: Thank you for this comment on the incidence-angle-dependent measurement. The samples in the *old* Fig. R6 and new Fig. R4 have different thicknesses. The thickness in the *old* Fig. R6 is only ~ 10 nm. Its THz intensity does not vary significantly with the incident angle when tilted by a small angle. Thus, a thicker sample with 35 nm was chosen for the incidence-angle-dependent measurement. We indeed observe the maximum THz amplitude under normal incidence in **Fig. R4** (see below).

Fig. R4 | Incidence-angle-dependent THz emission in PtTe₂ films with the thickness of 35 nm. To show the variation of the THz amplitude more clearly, we directly add the incident angles on the peaks in **b** to avoid misunderstanding.

Comment (3) *Related to point (2) above. The authors are correct to note that when the incident angle is reversed, the emitted THz does not change polarity, implying that it *cannot* be PDE. However, if the THz emission mechanism is not PDE, then one should expect that THz emission at normal incidence should be larger than at oblique incidence, which is *not* what the data shows. The authors did not address this point in their reply.*

Reply: Thank you for this comment. The result (see the above figure) does demonstrate that THz emission under normal incidence ($\alpha = 0^\circ$) is larger than that at oblique incidence. The detailed explanation on such an incidence-angle-dependent THz emission had been provided in **Supplementary Note 3** (Line 131-134 on Page 4).

Reviewer #2 (Remarks to the Author):

Authors replied to my comments well, and I found the manuscript is now revised satisfactorily. I therefore recommend the publication of this work in Nature Communications.

Reply: Thank you for your favorable recommendation for our work in *Nature Communications*.

Reviewers' Comments:

Reviewer #1:

None

RESPONSE TO REVIEWERS' COMMENTS (NCOMMS-24-07569A)

Only confidential comments to the Editor were submitted.

Reply: We thank Reviewer #1 for his/her comments that improve our manuscript and recommendation for publication in *Nature Communications*.